# PTC124 Rescues Nonsense Mutation of Two Tumor Suppressor Genes *NOTCH1* and *FAT1* to Repress HNSCC Cell Proliferation

**DOI:** 10.3390/biomedicines10112948

**Published:** 2022-11-16

**Authors:** Ming-Han Wu, Rui-Yu Lu, Si-Jie Yu, Yi-Zhen Tsai, Ying-Chen Lin, Zhi-Yu Bai, Ruo-Yu Liao, Yi-Chiang Hsu, Chia-Chi Chen, Bi-He Cai

**Affiliations:** 1School of Medicine, I-Shou University, No.8, Yida Rd., Jiaosu Village Yanchao District, Kaohsiung City 82445, Taiwan; 2Department of Medical Laboratory Science, I-Shou University, No.8, Yida Rd., Jiaosu Village Yanchao District, Kaohsiung City 82445, Taiwan; 3Department of Pathology, E-Da Hospital, No.1, Yida Rd., Jiaosu Village Yanchao District, Kaohsiung City 82445, Taiwan; 4College of Medicine, I-Shou University, No.8, Yida Rd., Jiaosu Village Yanchao District, Kaohsiung City 82445, Taiwan

**Keywords:** *NOTCH1*, *FAT1*, PTC124, nonsense mutation, HNSCC

## Abstract

(1) Background: PTC124 (Ataluren) is an investigational drug for the treatment of nonsense mutation-mediated genetic diseases. With the exception of the *TP53* tumor suppressor gene, there has been little research on cancers with nonsense mutation. By conducting a database search, we found that another two tumor suppressor genes, *NOTCH1* and *FAT1*, have a high nonsense mutation rate in head and neck squamous cell carcinoma (HNSCC). PTC124 may re-express the functional *NOTCH1* or *FAT1* in nonsense mutation *NOTCH1* or *FAT1* in HSNCC (2) Methods: DOK (with *NOTCH1* Y550X) or HO-1-u-1 (with *FAT1* E378X) HNSCC cells were treated with PTC124, and the *NOTCH1* or *FAT1* expression, cell viability, and *NOTCH1*- or *FAT1*-related downstream gene profiles were assayed. (3) Results: PTC124 was able to induce *NOTCH1* or *FAT1* expression in DOK and HO-1-u-1 cells. PTC124 was able to upregulate *NOTCH* downstream genes *HES5*, *AJUBA*, and *ADAM10* in DOK cells. PTC124 enhanced DDIT4, which is under the control of the *FAT1*–YAP1 pathway, in HO-1-u-1 cells. FLI-06 (a NOTCH signaling inhibitor) reversed PTC124-mediated cell growth inhibition in DOK cells. PTC124 could reverse TT-10 (a YAP signaling activator)-mediated HO-1-u-1 cell proliferation. (4) Conclusions: PTC124 can rescue nonsense mutation of *NOTCH1* and *FAT1* to repress HNSCC cell proliferation.

## 1. Introduction

Nonsense mutation is a point mutation within the coding region that introduces a pre-stop codon (TGA or TAG or TAA) that produces truncated proteins. Nonsense mutation(s) can produce premature termination codons (PTCs), which lead to nonsense-mediated mRNA decay (NMD) with almost no expression of full-length proteins and fairly rare expression of truncated proteins [1]. Aminoglycoside drugs, such as G418 and gentamicin, cannot only act as antibiotics to target the bacterial ribosome but can also inhibit NMD and promote PTC readthrough to rescue full-length proteins [2,3]. However, these aminoglycoside drugs have high toxicity, which limits their clinical use for nonsense mutation-mediated diseases [4]. Furthermore, aminoglycosides may generate free radicals that damage sensory cells and neurons within the inner ear and cause permanent hearing loss [5]. One non-aminoglycoside drug, PTC124 (Ataluren), has also been found to increase the readthrough ability of PTC without the ability to inhibit NMD [6]. PTC124 has low toxicity compared to gentamicin [7] and is also well-tolerated by patients [8,9,10]. PTC124 has been used in clinical phase II and III trials to treat genetic diseases characterized by specific nonsense mutations such as cystic fibrosis and Duchenne muscular dystrophy [11,12,13,14]. PTC124 has received conditional approval by the European Medicines Agency as an orally administered drug for the treatment of nonsense mutation-mediated Duchenne muscular dystrophy [15]; however, the United States Food and Drug Administration (FDA) has still not approved the drug for the treatment of this disease. PTC124 has been used in clinical phase I and II trials to treat metastatic deficient mismatch repair (dMMR) and mismatch repair proficient (pMMR) colorectal carcinoma or metastatic dMMR endometrial carcinoma to promote translation of additional out-of-frame code to enhance the effect of Pembrolizumab anti-PD1 therapy [16]. So, PTC124 should also have high value for clinical application for other cancer treatments of nonsense-mediated cancer formation.

*TP53* is a well-known tumor suppressor gene with a high mutation rate in cancers [17,18,19]. *TP53* has a 10.71% nonsense mutation rate among all the cancer mutation samples in the COSMIC database [20,21]. On the other hand, *TP53* has a 11.4% nonsense mutation rate in head and neck squamous cell carcinoma (HNSCC) mutation samples in the COSMIC database (Figure 1A) [21]. PTC124 is able to promote readthrough of nonsense mutations in *TP53* to produce functional full-length p53 [22]. In this study, we conducted a database search and found that two other tumor suppressor genes, *NOTCH1* and *FAT1*, have a high nonsense mutation rate (*NOTCH1*: 19.62% and *FAT1*: 40.11%) in HNSCC mutation samples in the COSMIC database compared to other cancers (Figure 1B,C) [21]. Compared to p53, which only has 393 amino acid residues with a molecular weight of 53 kDa [23], *NOTCH1* and *FAT1* have 2555 and 4588 amino acid residues with molecular weights of 273 and 506 kDa, respectively [24,25].

*NOTCH1* and *FAT1* are both transmembrane proteins. The *NOTCH1* ligands are Delta-like-1 and Jagged-1 [26]. Ligand binding can lead *NOTCH1* to cleave and release the Notch intracellular domain (NICD) [27]. NICD can translocate to the nucleus and associate with CSL, MAML, and SKIP protein complex to active HES family and NOTCH target genes [27]. On the other head, dachsous (ds) is known to function as an *FAT1* ligand in Drosophila [28], but no *FAT1* ligand had been identified in mammals [25,28]. *FAT1* was shown to regulate Hippo kinase components as a scaffold for Hippo kinases to enhance their kinase activation [29]. Loss of *FAT1* leads to dismantling of the Hippo core complex and YAP dephosphorylation, causing YAP translocation to the nucleus [29]. High expression of *NOTCH1* correlated with better overall and disease-free survival in HNSCC patients [30]. Young-onset head and neck cancer patients with *FAT1* germline variants had a shorter overall survival [31]; therefore, restoration of the functional *NOTCH1* or *FAT1* expression in nonsense mutation *NOTCH1* or *FAT1* in HSNCC is a key issue in anticancer therapy. In this study, we treated two HNSCC cell lines with nonsense mutations of *NOTCH1* and *FAT1* with PTC124 and measured the change in the cell proliferation of these two tumor suppressor genes.

## 2. Materials and Methods

### 2.1. Cell Culture and Drug Treatment

The HNSCC cell lines SAS, DOK, and HO-1-u-1 were maintained at 37 °C in 5% CO_2_ in DMEM (for SAS and DOK) with 5 µg/mL hydrocortisone or DMEM/F12 (for HO-1-u-1) (Invitrogen, Carlsbad, CA, USA), supplemented with 10% FBS (Invitrogen), 100 U/mL penicillin, and 100 μg/mL streptomycin (both from Invitrogen). HNSCC cells were treated for 48 h with mock (DMSO only), 50 μM PTC124 (MedChemExpress, Monmouth Junction, NJ, USA), 10 μM FLI-06 (MedChemExpress, Monmouth Junction, NJ, USA), or 10 μM TT-10 (AOBIOUS, Gloucester, MA, USA).

### 2.2. Immunocytochemistry

The culture medium was first removed from each well and then washed twice with 1X PBS. Formaldehyde fixative solution (100 μL; 4% paraformaldehyde in 1X PBS) was added to each well and incubated for 20 min. Wash buffer (100 μL; 1% BSA in 1X PBS) was used to wash each well twice. Blocking buffer (100 μL; 1% BSA, 0.2% Triton X-100 in 1X PBS) was added and then the mixture was incubated for 30 min. After the blocking buffer was removed, the primary antibody p53 (1 μL; Santa Cruz; DO-7 sc-47698), *NOTCH1* (1 μL; Elabscience; E-AB-12815) or *FAT1* (1 μL; Elabscience; E-AB-13237) (1:100) was prepared in 100 µL blocking buffer to stain cells for 120 min. After staining, each well was washed three times with 100 µL wash buffer. Hoechst 33,342 stock solution (1 mg/mL) and secondary antibody PE-conjugated mouse IgG or PE-conjugated rabbit IgG (0.5 µL; 1:200) were added to 100 µL blocking buffer to stain each well for 30 min in the dark. The wells were washed three times with 100 µL wash buffer. Finally, the images were visualized and captured using an inverted fluorescence microscope (ECLIPSE. Ts2; Nikon, Tokyo, Japan). The Mean Gray Value was used to quantify the fluorescence intensity using ImageJ [32].

### 2.3. Real-Time RT-PCR

Total cellular RNA was extracted with TRIzol reagent (Invitrogen). Total RNA (1 μg) was converted to first-strand cDNA using the QuantiNova Reverse Transcription Kit (QIAGEN, Hilden, Germany). cDNA samples were mixed with 5X HOT FIREPol EvaGreen qPCR Mix Plus (Omics Bio, New Taipei City, Taiwan) and primers. The amplification of each qPCR reaction was monitored by the QuantStudio 3 Real-Time PCR System (Applied Biosystems). The relative expression of the transcript levels was calculated as 2−ΔΔCT. Each pair of forward (F) and reverse (R) primers was as follows: *HES5*, F: AAGCACAGCAAAGCCTTCGT and R: CTGCAGGCACCACGAGTAG; *AJUBA*, F: CGTTGCAAGGCTTTCTACAGT and R: ACAATGCATCGGAAACAGCC; *ADAM10*, F: ACGGAACACGAGAAGCTGTG and R: CGGAGAAGTCTGTGGTCTGG; *DDIT4*, F: TACCTGGATGGGGTGTCGTT and R: ACCAACTGGCTAGGCATCAG; *CTGF*, F: GTTTGGCCCAGACCCAACTA and R: GGCTCTGCTTCTCTAGCCTG; *GAPDH*, F: GTCTCCTCTGACTTCAACAGCG and R: ACCACCCTGTTGCTGTAGCCAA.

### 2.4. Reporter Constructs and Luciferase Assays

The 2X *NOTCH1*/HES consensus sequence or 2X YAP1/TEAD consensus sequence constructs containing two repeats of CGTGGGAA or ACATTCCA were cloned into the SmaI site of the pGL3 promoter as previously described [33]. We co-transfected 1 μg pGL3-2X *NOTCH1*/HES firefly luciferase plasmid or pGL3-2X YAP1/TEAD firefly luciferase plasmid and 10 ng pRL-pCMV Renilla luciferase plasmid (Promega, Madison, WI, USA) into cells using 3 μL TransIT-X2 transfection reagent (Mirus Bio, Madison, WI, USA) with 60% confluence in each well of a 12-well tissue culture dish. After 8 h of transfection, cells were treated with DMSO or 50 μM PTC124 for 2 days. The cells in each well were washed three times with 1 mL 1X PBS. Cells were harvested in 0.25 mL reporter lysis buffer and subjected to a dual luciferase assay (Promega, Madison, WI, USA) according to the manufacturer’s protocol. Firefly luciferase activity was normalized to Renilla luciferase activity, and the data are presented as the mean ± standard deviation from three independent experiments, each of which was performed in triplicate.

### 2.5. CCK8 Assay

CCK8 assay (Invitrogen) was used to assess the cell viability. CCK8 (10 μL) was added to each well of a 96-well plate, and the plate was placed in a 37 °C incubator with 5% CO_2_ for 1 h. After incubation, the OD 450 nm was read on a SpectraMax iD3 microplate reader (Molecular Devices, Silicon Valley, CA, USA). Mock was calculated as 100% to normalize for other drug treatment conditions. Medium without cells was used as the blank, and the percentage cell viability was calculated as follows: (Drug OD-Blank OD/Mock OD-Blank OD) × 100%.

### 2.6. Statistical Analysis

The statistical differences between the two groups were assessed by the Student’s *t* test (two-tailed). All results are presented as the mean ± SD. A P value of less than 0.05 was considered statistically significant (* *p* < 0.05, ** *p* < 0.01, *** *p* < 0.001).

## 3. Results

According to the gene mutation rate in HNSCC from the COSMIC database [21] (all data were retrieved on 1 October 2022), *TP53* has a similar nonsense mutation rate in HNSCC as in other types of cancers (Figure 1A). Compared to the other types of cancers, *NOTCH1* and *FAT1* have a relatively high nonsense mutation rate in head and neck squamous cell carcinoma (HNSCC) (Figure 1B,C). In this study, we searched the COSMIC Cell Lines Project to find the suitable cells with *TP53* or *NOTCH1* or *FAT1* nonsense mutation cells (Table 1) [21]. An HNSCC cell line, SAS, contains TAG-type nonsense mutation in the coding region of *TP53* (Table 1). We also used two HNSCC cell lines, DOK and HO-1-u-1 cells, which both contain TAA-type nonsense mutation in the coding region of *NOTCH1* and *FAT1*, respectively (Table 1).

PTC124 can induce p53 expression in SAS cells (Figure 2A). We found that PTC124 can induce *NOTCH1* expression in DOK cells (Figure 2B), and PTC124 can induce *FAT1* expression in HO-1-u-1 cells (Figure 2C). Three *NOTCH1* downstream genes, *HES5*, *AJUBA*, and *ADAM10*, could be induced by PTC124 in DOK cells (Figure 3A). *AJUBA* and *ADAM10* also act as tumor suppressor genes in HNSCC [34]. *FAT1* can repress the YAP1 signal [35], and *DDIT4* is a negative YAP1 target gene [36]. DDIT4 could be induced by PTC124 in HO-1-u-1 cells (Figure 3B). *CTGF* is a YAP1 downstream gene [37]; PTC124 can repress CTGF expression in HO-1-u-1 cells (Figure 3B).

To assay whether PTC124 can modulate the transactivation function of HES and YAP, we cloned the 2X *NOTCH1*/HES consensus sequence firefly luciferase reporter, which contains the repeat CGTGGGAA, and the 2X YAP1/TEAD consensus sequence firefly luciferase reporter, which contains a ACATTCCA repeat. PTC124 can stabilize firefly luciferase directly [38], but washing the cells three times before adding the reporter assay lysis buffer can remove the residual PTC124 to influence firefly luciferase [39]. So, we followed the same washing protocol to perform the reporter assay. PTC124 can activate 2X *NOTCH1*/HES consensus sequence reporter assay in DOK cells (Figure 4A). PTC124 can repress 2X YAP1/TEAD consensus sequence reporter assay in HO-1-u-1 cells (Figure 4B).

One NOTCH signaling inhibitor, FLI-06, was able to reverse PTC124-mediated cell growth inhibition in DOK cells (Figure 5A). A YAP signaling activator, TT-10, was able to promote HO-1-u-1 cell growth (Figure 5B). PTC124 was able to reverse TT-10-mediated HO-1-u-1 cell proliferation (Figure 5B). Therefore, PTC124 was able to rescue nonsense mutation of *NOTCH1* and *FAT1* to repress HNSCC cell proliferation (Figure 6).

## 4. Discussion

PTC124 is considered a relatively safe drug for nonsense mutation treatment compared with aminoglycosides G418 and gentamicin [7], but PTC124 generated fewer readthrough products than the aminoglycosides [40]. One study used a low dosage of G418 or gentamicin and PTC124 co-treatment to stimulate readthrough efficiency [41]. Several other drugs have also been tested together with a low dosage of G418 to try to generate more full-length PTC readthrough products, such as CDX5-1, mefloquine, SMG1i, Y-320, CC885, CC-90009, SRI-37240, and SRI-41315 [42,43,44,45,46,47,48]. SMG1i showed no response in enhancing PTC124 PTC readthrough [44], but CDX5-1, mefloquine, Y-320, CC885, CC-90009, SRI-37240, and SRI-41315 have not been tested with PTC124 as nonsense mutation treatment [42,43,45,46,47,48]. Whether there are any non-aminoglycoside drug(s) that can promote PTC124 PTC readthrough is an important consideration in the quest to improve the clinical use of PTC124 for nonsense mutation-mediated genetic diseases and cancers. Of note, one interesting study found that caffeine can attenuate the NMD activity and combine with PTC124 to enhance the readthrough and enrich the generation of full-length proteins [49]. Many small compounds have been screened and identified to have PTC readthrough activity through cell-based luciferase assays with a premature nonsense mutation in a firefly luciferase gene [2,47,50]. However, PTC124 can stabilize firefly luciferase directly [38], so this may restrict the identification of small compound(s) to enhance the PTC124 PTC readthrough. PTC124 has an influence on Renilla reniformis luciferase [51]. This novel type of reporter assay system is also a powerful tool for screening potential small compound(s) to enhance the PTC124 PTC readthrough. 

In addition, one PTC124 analogue, NV848, was also found to be able to promote K62X SBDS gene nonsense mutation readthrough [52]. NV848 has also been tested with low toxicity in zebrafish [52]. The authors also mentioned that NV848 is more hydrophilic and should provide much easier delivery than PTC124 [52]. NV848 may also be very useful to rescue nonsense mutation of key tumor suppressor genes such as *TP53*, *NOTCH1*, and *FAT1*. Caffeine can enhance PTC124 readthrough [39], so NV848 may also have a good response with caffeine to rescue nonsense mutation of key tumor suppressor genes. Another two PTC124 analogs, NV914 and NV930, can also restore a much greater amount of CFTR W1282X protein than NV848 and PTC124 [53]. These two PTC124 analogs may also have a good response to rescue the nonsense mutation of key tumor suppressor genes.

The rates of pathological nonsense mutations occurring as three types of stop codons are TGA (38.5%), TAG (40.4%), and TAA (21.1%) [54]. Sixty-one codon triplets encode the twenty different types of amino acids, but only eighteen triplets encoding ten different types of amino acids can be converted to a stop codon through a single-base pair substitution mutation. These eighteen triplets are CGA or AGA (encode Arg), TGT or TGC (encode Cys), CAG or CAA (encode Gln), GAG or GAA (encode Glu), GGA (encode Gly), TTG or TTA (encode Leu), AAG or AAA (encode Lys), TCG or TCA (encode Ser), TGG (encode Trp), and TAT or TAC (encode Tyr). The most frequent types of mutation of nucleotide substitutions that become nonsense mutations are CGA to TGA (21%) and CAG to TAG (19%) [54]. The efficiency of aminoglycosides and PTC124 for PTC readthrough of DNA pre-stop codons is TGA > TAG > TAA [6,55]. 2,6-Diaminopurine (DAP) has a PTC readthrough effect, but DAP is only effective for TGA but not TAA or TAG [56]. The nucleoside analog, clitocine, has PTC readthrough efficiency TAA > TGA > TAG [57]. DOK (with *NOTCH1* Y550X; single-base pair substitution mutation from TAC to TAA) or HO-1-u-1 (with *FAT1* E378X; single-base pair substitution mutation from GAA to TAA), which were used in this study, are both TAA-type nonsense mutations (Table 1). Therefore, clitocine may be a powerful drug to induce more *NOTCH1* or *FAT1* expression in our cell system. 

According to our results, PTC124 can restore *NOTCH1* expression to enhance the HES transactivation function in DOK cells (Figure 2C and Figure 4A). PTC124-mediated cell growth repression was reversed by FLI-06 (a NOTCH signaling inhibitor) in DOK cells (Figure 5A). Overexpression of wild-type *NOTCH1* in *NOTCH1* mutant HNSCC cell lines can rapidly decrease cell viability and proliferation [58]; therefore, re-expressing functional *NOTCH1* in both nonsense and missense mutations of *NOTCH1* is a good anticancer strategy in HNSCC. PTC124 can restore *FAT1* expression to block the YAP1 transactivation function in HO-1-u-1 cells (Figure 2E and Figure 4B), and PTC124 can upregulate the expression of the key tumor suppressor gene DDIT4 (Figure 3B). Cisplatin can induce several tumor suppressor genes such as p53, p73, p21, and DDIT4 but can also repress YAP1 in HNSCC [42]. Recently, DDIT4 has also been reported to be indirectly activated by p53 through RFX7 regulatory factor X 7 (RFX7) [43]. The combination treatment of PTC124 and chemotherapy drugs such as Cisplatin may be a good method of tumor therapy in YAP1 nonsense mutation HNSCC. This study demonstrated that PTC124 has an anticancer effect in several key tumor suppressor genes with nonsense mutation in HNSCC. The limitation of potential usage of this strategy may be the frequency of nonsense mutation of each key tumor suppressor gene in different types of cancers. Further experiments are needed to further investigate the pre-clinical and clinical usage of PTC124 for anticancer purposes in nonsense-mediated cancer formation. 

## 5. Conclusions

Two tumor suppressor genes *NOTCH1* and *FAT1* both have a high nonsense mutation rate in HNSCC. PTC124 can rescue nonsense mutation of *NOTCH1* and *FAT1* to repress cell proliferation in HNSCC. PTC124 and other PTC124 analogues may also be effective on other tumor suppressor genes with high nonsense mutation rates in other types of cancers.

## Figures and Tables

**Figure 1 biomedicines-10-02948-f001:**
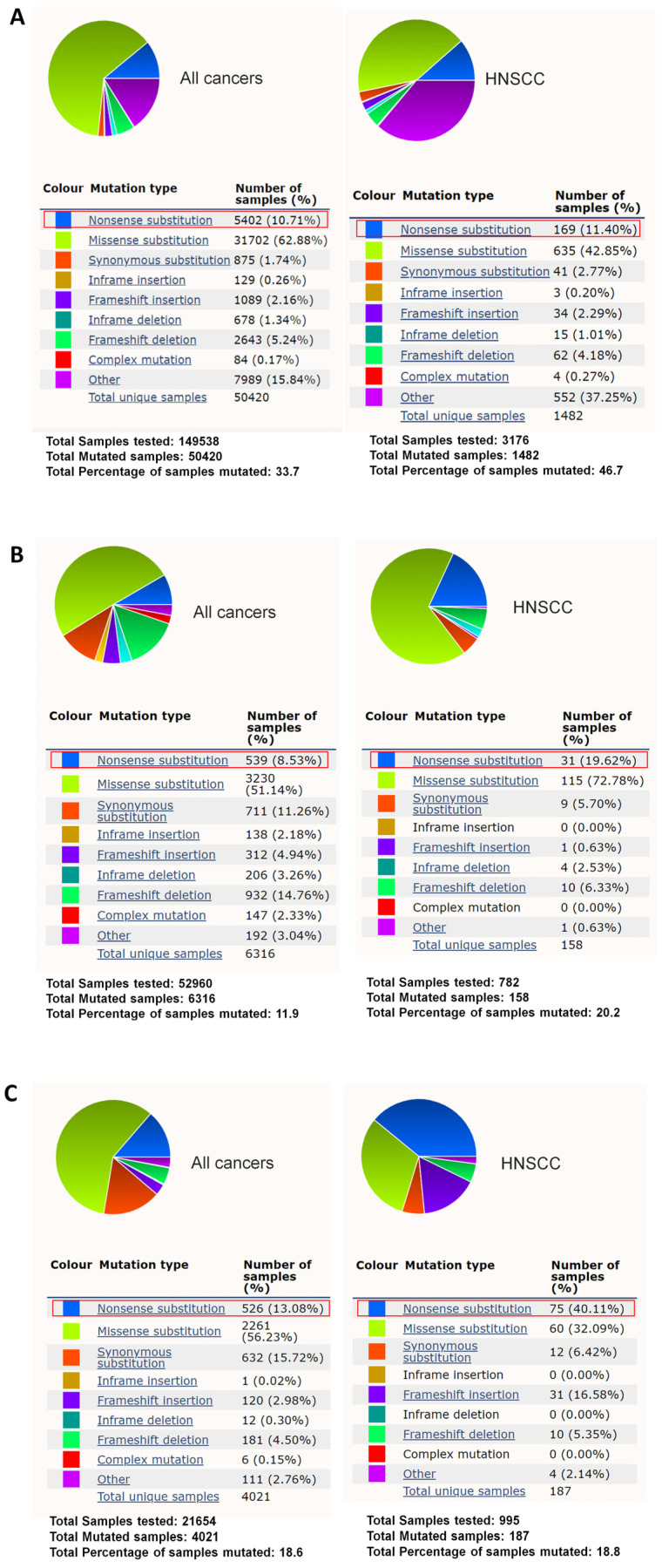
*NOTCH1* and *FAT1* have a high nonsense mutation rate in HNSCC. (**A**) *TP53* has a 10.71% nonsense mutation rate in all the cancer mutation samples and a 11.4% nonsense mutation rate in HNSCC with *TP53* mutation in the COSMIC database (https://cancer.sanger.ac.uk/cosmic (accessed on 1 October 2022)) [21]. (**B**) *NOTCH1* only has an 8.53% nonsense mutation rate across all the cancer mutation samples but a 19.62% nonsense mutation rate in HNSCC with *NOTCH1* mutation in the COSMIC database [21]. (**C**) *FAT1* only has a 13.08% nonsense mutation rate across all the cancer mutation samples but a 40.11% nonsense mutation rate in HNSCC with *FAT1* mutation in the COSMIC database [21].

**Figure 2 biomedicines-10-02948-f002:**
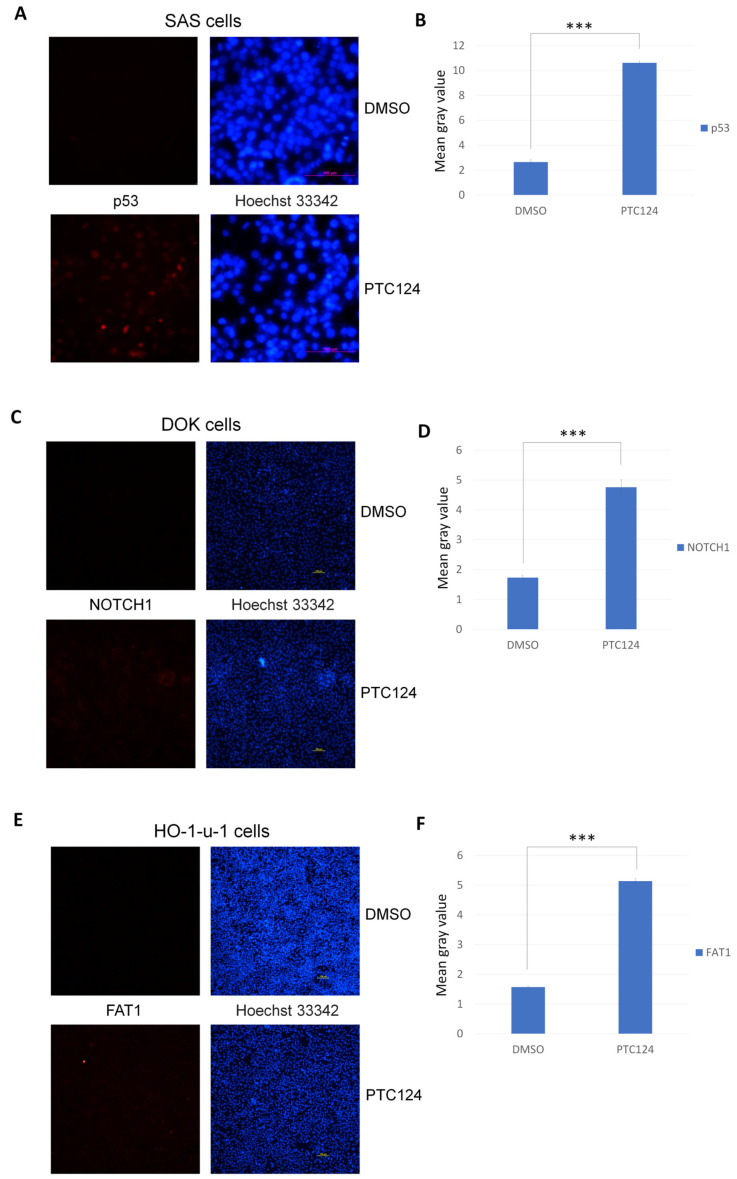
PTC124 can induce *NOTCH1* and *FAT1* expression in DOK and HO-1-u-1 cells, respectively. (**A**) PTC124 (50 μM) can induce p53 in SAS cells. (**B**) p53 fluorescence intensity is dramatically inducted by PTC124 (*** *p* < 0.001). Results are displayed as mean ± SD, *n* = 3. (**C**) PTC124 (50 μM) can induce *NOTCH1* in DOK cells. (**D**) *NOTCH1* fluorescence intensity is dramatically induced by PTC124 (*** *p* < 0.001). Results are displayed as mean ± SD, *n* = 3. (**E**) PTC124 (50 μM) can induce *FAT1* in HO-1-u-1 cells. (**F**) *FAT1* fluorescence intensity is dramatically induced by PTC124 (*** *p* < 0.001). Results are displayed as mean ± SD, *n* = 3.

**Figure 3 biomedicines-10-02948-f003:**
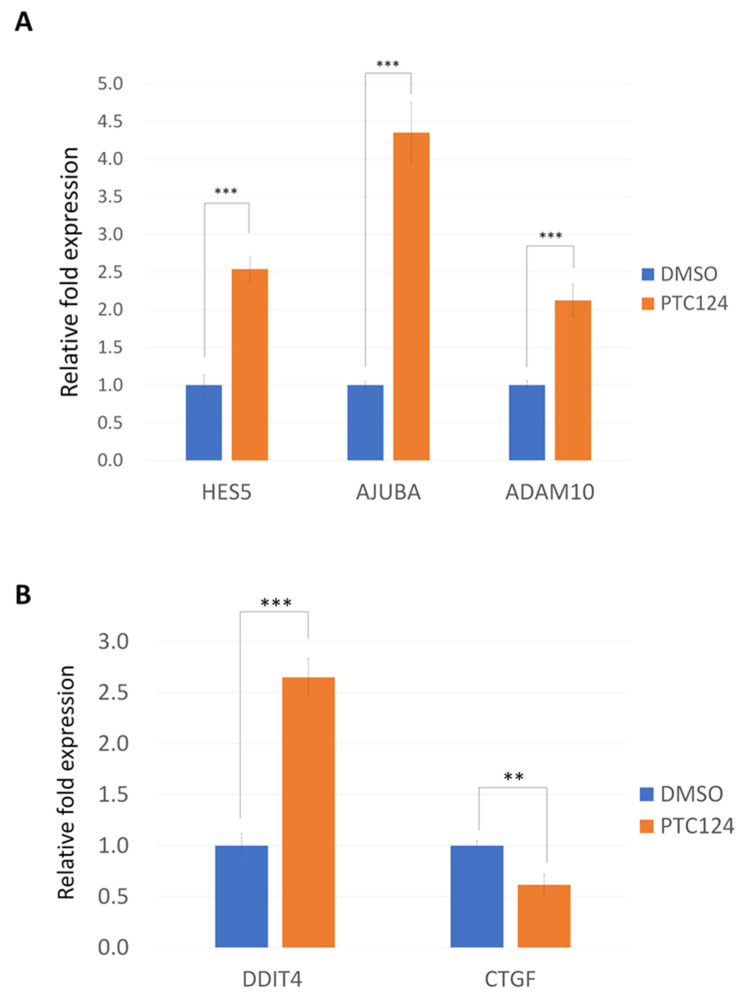
PTC124 can induce *NOTCH1* downstream genes and DDIT4 in DOK and HO-1-u-1 cells, respectively. (**A**) PTC124 (50 μM) can induce HES5, AJUBA, and ADAM10 expression in DOK cells (*** *p* < 0.001). DMSO only was calculated as 1 to normalize for the other conditions. Results are displayed as mean ± SD, *n* = 3. (**B**) PTC124 (50 μM) can induce DDIT4 and repress CFGF expression in HO-1-u-1 cells (** *p* < 0.01, *** *p* < 0.001). DMSO only was calculated as 1 to normalize for the other conditions. Results are displayed as mean ± SD, *n* = 3.

**Figure 4 biomedicines-10-02948-f004:**
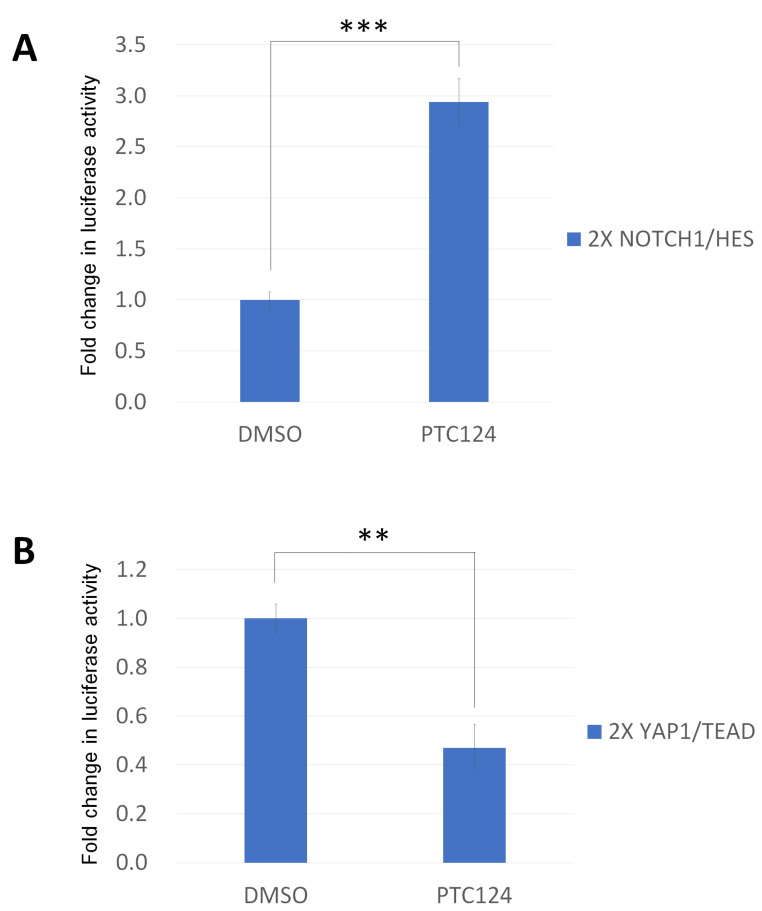
PTC124 can moderate HES and TEAD transactivation function. (**A**) PTC124 (50 μM) can upregulate 2X *NOTCH1*/HES consensus sequence reporter assay in DOK cells (*** *p* < 0.001). DMSO only was calculated as 1 to normalize for the other conditions. The results are displayed as mean ± SD, *n* = 3. (**B**) PTC124 (50 μM) can downregulate 2X YAP1/TEAD consensus sequence reporter assay in HO-1-u-1 cells (** *p* < 0.01). DMSO only was calculated as 1 to normalize for the other conditions. Results are displayed as mean ± SD, *n* = 3.

**Figure 5 biomedicines-10-02948-f005:**
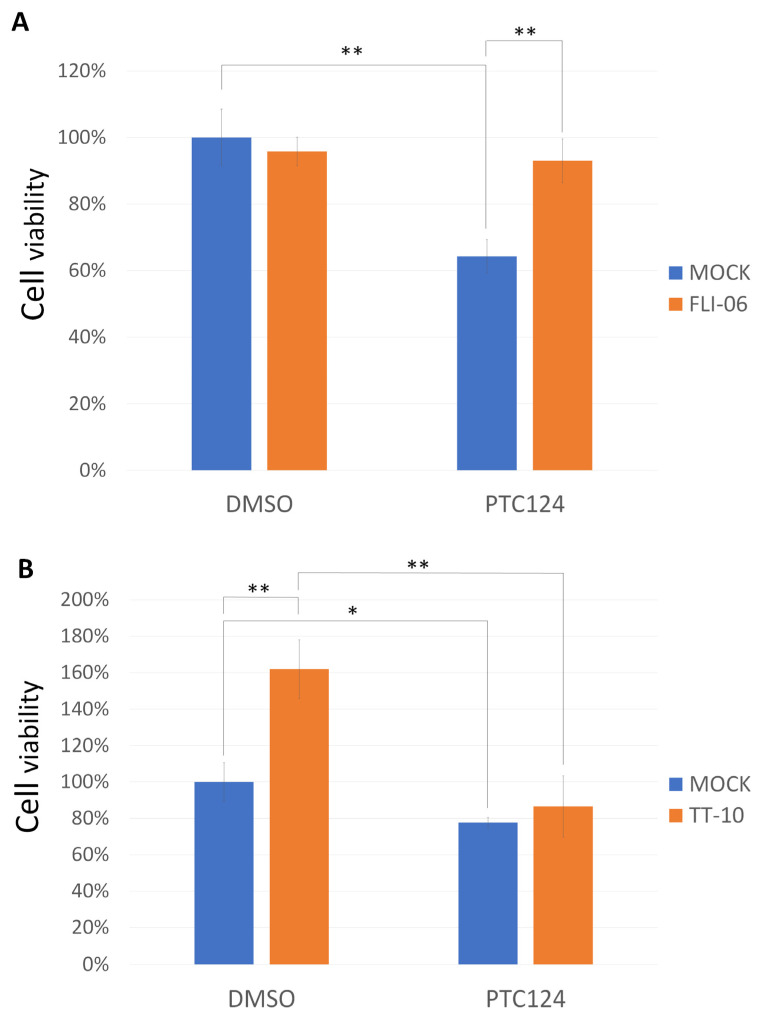
PTC124 can control HNSCC cell growth in DOK and HO-1-u-1 cells. (**A**) FLI-06 (10 μM) was able to reverse PTC124-mediated repression of DOK cell growth (** *p* < 0.01). DMSO/MOCK was calculated as 1 to normalize for the other conditions. Results are displayed as mean ± SD, *n* = 3. (**B**) PTC124 could repress HO-1-u-1 cell growth (* *p* < 0.05). PTC124 could reverse TT-10 (10 μM)-mediated HO-1-u-1 cell proliferation (** *p* < 0.01). DMSO/MOCK was calculated as 1 to normalize for the other conditions. Results are displayed as mean ± SD, *n* = 3.

**Figure 6 biomedicines-10-02948-f006:**
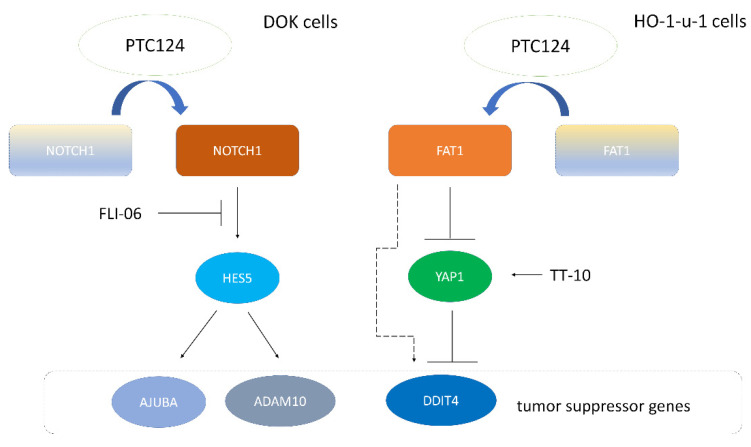
PTC124 can rescue nonsense mutation of two tumor suppressor genes *NOTCH1* and *FAT1* to repress HNSCC cell proliferation. PTC124 can induce the expression of *NOTCH1* (left blue arrow) and *NOTCH1* downstream genes (*HES5*, *AJUBA*, and *ADAM10*) in DOK cells (with *NOTCH1* Y550X). FLI-06 is a NOTCH signaling inhibitor. FLI-06 can reverse PTC124-mediated cell growth inhibition in DOK cells. PTC124 can induce *FAT1* (right blue arrow) and DDIT4 (dash line arrow) expression in HO-1-u-1 cells (with *FAT1* E378X). TT-10 is a YAP signaling activator. PTC124 can reverse TT-10-mediated HO-1-u-1 cell proliferation.

**Table 1 biomedicines-10-02948-t001:** Information about the cell lines used in this study.

Cell Line Name	Nonsense Mutation Gene	Protein Mutation Site *	DNA Mutation Site *	Original Codon	Mutated Pre-Stop Codon
SAS	*TP53*	p.E336X	c.1006G > T	GAG	TAG
DOK	*NOTCH1*	p.Y550X	c.1650C > A	TAC	TAA
HO-1-u-1	*FAT1*	p.E378X	c.1132G > T	GAA	TAA

* The protein mutation site and DNA mutation site of each mutation in the cell line are available from the COSMIC Cell Lines Project (https://cancer.sanger.ac.uk/cell_lines (accessed on 1 October 2022) [21].

## Data Availability

Data can be provided by the corresponding author upon reasonable request. Data were obtained from the COSMIC database, which is freely available for non-commercial users.

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
