# Peer review of "PTC124 Rescues Nonsense Mutation of Two Tumor Suppressor Genes NOTCH1 and FAT1 to Repress HNSCC Cell Proliferation"

_biomedicines, 2022, doi:10.3390/biomedicines10112948_

Round 1

Reviewer 1 Report

The presented paper investigates the effect of ataluren on the induction of the expression of NOTCH1 and FAT1 by reading through the nonsense mutations that lead to the inactivation of these genes in DOK and HO-1-u-1 cells. The paper is metodologically sound and the Authors conclude that ataluren increases the expression of NOTCH1 and FAT1 which affects cell proliferation. The paper is interesting but several issues need to be addressed before publication (see below).

Please, write gene names in italics throughout the manuscript (including the title).

Line 35 - TAA codon is mentioned twice (should be TGA in the second case)

Introduction: Instead of describing nonsense mutation-driven non-cancer diseases (although it is very interesting in itself), it would be desirable to indicate the significance of NOTCH and FAT1/YAP modulation as therapeutic strategies in HNSCC, in order to better justify the aim of the study.

Paragraph 2.1: how long were drug treatments?

Lines 114-115 - "Total RNA (ul)..." - should it not be "(ug)"?

Line 137: "head and neck squamous cell carcinoma (HNSCC)" - please, explain the abbreviation at its first appearance in the manuscript (it was used earlier in the text of the manuscript)

Lines 135-140 - a short sentence regarding the way of acquisition of the presented COSMIC database results would be desirable

Lines 170-173 (starting with "Cisplatin...") - please transfer this information to the discussion section and stick to the description of the obtained results in the Results section

Figure 2 - could you also provide graphs with quantitative results (with statistics)? The induction of the expression of the proteins seems very modest, however this may be rtelated to the quality of the images. Quantitative data would be more objective.

Line 181 - please, use "respectively" instead of "separately"

Lines 183-184 - the description of the genes in not necessary in the caption, I would suggest deleting it.

Figures 4 - what about statistical analysis? Were the differences statistically significant?

It would be desirable to analyze the expression of another YAP target gene: CTGF.

Line 211 - "...full length PTC..." what do the Authors mean?

Discussion - in its current form the discussion does not discuss the described results at all. Therefore, the whole discussion has to be rewritten in order to address the meaning of the presented results with regard to the potential of using PTC124 in the treatment of HNSCC. The discussion should refer directly to study hypothesis, which has to be clearly stated in the last part of the introduction (currently it is missing).

In order to finally prove that PTC124 acts via the modulation of the transcriptional activity of NOTCH/NICD and YAP, it would be desirable to perform additional experiments in order to check the binding of the proteins to DNA and the induction of reporter genes (if possible).

Reviewer 2 Report

The reviewed study on Atularen as an agent rescuing  nonsense mutations in some genes  is a piece of good molecular work. An attractive  therapeutic perspective of Atularent application in HNSCC has been only slightly discussed.

I would like to suggest to consider the following points, potentially contributing further to the paper quality.

1. As the study deals with HNSCC tumors that means laryngology/foniatry clinicians and researchers are potential readers  I would say openly about hearing loss following application of aminoglycoside drugs.

2. I wonder what is a difference (characteristics) between two used cell lines. However, I found the information later in the text  (Table 1.).

3.  A single dose of PTC124 and FLI-06 was used to treat cell lines. Was the dose derived from pilot non-described experiments or deduced from clinical application? This information should be given in the text.

4. p53 mutation were mentioned several times. Why p53 cell line was not included as an imposing control? I realize the study is alreadyompleted and I am not going to perform such extra experiments.

Reviewer 3 Report

The authors aimed, to treated two HNSCC cell lines with nonsense mutations of NOTCH1 and FAT1 with PTC124 and measured the change in cell proliferation of these two tumor  suppressor genes.

The study covers some issues that have been overlooked in other similar topics. The structure of the manuscript appears adequate and well divided in the sections. Moreover, the study is easy to follow, but some issues should be improved. Some of the comments that would improve the overall quality of the study are:

a. Authors must pay attention to the technical terms acronyms they used in the text.

b. Better stated the aim of the study in the abstract and introduction section.

b. Limitations of the study needs to be added.

c. Conclusion Section: This paragraph required a general revision to eliminate redundant sentences and to add some "take-home message".

Round 2

Reviewer 1 Report

The Authors did an excellent job and have addressed all the raised issues. Therefore, the paper is almost ready for publication. Below are a few minor points that I feel need to be addressed to finally improve the paper.

I would suggest deleting lines 54-67, since this information does not directly refer to the focus of the study, which is the use of ataluren in HNSCC cells. Instead, I would suggest adding one sentence regarding the attempts of using ataluren in cancer treatment, in cases with driver nonsense mutations.

Please, make sure that all gene names are written in italics (e.g. lines 164-171: NOTCH1, FAT1, TP53). In fact, the gene name for p53 is TP53.

Paragraph 2.4 - please, provide the name of the transfection agent.

Overall, congratulations for your excellent paper.
